# CRISPR-Induced Expression of N-Terminally Truncated Dicer in Mouse Cells

**DOI:** 10.3390/genes12040540

**Published:** 2021-04-08

**Authors:** Radek Malik, Petr Svoboda

**Affiliations:** Institute of Molecular Genetics of the Czech Academy of Sciences, Videnska 1083, 142 20 Prague 4, Czech Republic; malikr@img.cas.cz

**Keywords:** RNAi, Dicer, CRISPR, dCas9, VP64, MS2, sgRNA

## Abstract

RNA interference (RNAi) designates sequence-specific mRNA degradation mediated by small RNAs generated from long double-stranded RNA (dsRNA) by RNase III Dicer. RNAi appears inactive in mammalian cells except for mouse oocytes, where high RNAi activity exists because of an N-terminally truncated Dicer isoform, denoted Dicer^O^. Dicer^O^ processes dsRNA into small RNAs more efficiently than the full-length Dicer expressed in somatic cells. Dicer^O^ is expressed from an oocyte-specific promoter of retrotransposon origin, which is silenced in other cell types. In this work, we evaluated CRISPR-based strategies for epigenetic targeting of the endogenous Dicer gene to restore Dicer^O^ expression and, consequently, RNAi. We show that reactivation of Dicer^O^ expression can be achieved in mouse embryonic stem cells, but it is not sufficient to establish a robust canonical RNAi response.

## 1. Introduction

Canonical RNA interference (RNAi) has been defined as sequence-specific RNA degradation induced by long double-stranded RNA (dsRNA) [1]. RNAi is initiated by processing long dsRNA by RNase III Dicer into ~22 nt small interfering RNAs (siRNAs), which guide recognition and endonucleolytic cleavage of complementary mRNA molecules (reviewed in [2]). The mammalian canonical endogenous RNAi pathway is generally weak, if active at all (reviewed in [3]). Although mammalian genomes encode protein factors necessary and sufficient for reconstituting canonical RNAi in vitro [4] or in the yeast [5,6], these protein factors primarily support a gene-regulating microRNA pathway where small RNAs are produced from genome-encoded small hairpin precursors and typically guide translational repression coupled with transcript destabilization (reviewed in [7]). Furthermore, the primary mammalian mechanism responding to dsRNA in somatic cells is not RNAi but a sequence-independent interferon pathway (reviewed in [8]). The interferon pathway is an essential component of mammalian innate immunity and one of the factors impeding RNAi [9,10,11,12]. Another key factor limiting mammalian RNAi is low Dicer activity, which can be enhanced by truncating the Dicer’s N-terminal helicase domain [9,12,13,14]. Mouse oocytes, the only known mammalian cell type where RNAi is highly active and functionally important, express a unique, naturally N-terminally truncated Dicer isoform, denoted Dicer^O^ (Figure 1A) [14]. This truncated isoform arose upon intronic insertion of a long terminal repeat (LTR) from the MTC retrotransposon subfamily, which provides an oocyte-specific promoter and the first exon of Dicer^O^ (Figure 1B). This isoform is expressed only in mouse oocytes, and it was not observed in transcriptomes of somatic cells [14]. Previous studies showed that functional canonical RNAi could be restored in mammalian somatic cells under specific conditions [9,10,12,14], but it was unclear whether Dicer^O^ reexpression from the endogenous *Dicer* locus could achieve such an effect as well. Accordingly, we examined whether Dicer^O^ expression could be induced from the endogenous *Dicer* locus in mouse cells and whether Dicer^O^ reactivation would be sufficient to restore robust endogenous canonical RNAi.

As Dicer^O^ expression is restricted to oocytes and transcription factor(s) controlling it are unknown, we opted for an artificially designed transcriptional activator. Advances in the development of guided nucleases (reviewed in [16]) brought an opportunity to modify these sequence-specific enzymes as platforms for modulating mammalian transcription by fusing them with transcription activation or repression domains [17,18]. Among the guided nucleases, the clustered regularly interspaced short palindromic repeat (CRISPR) Cas9 nuclease gained major popularity because of simple programming of its sequence specificity with a single-guide RNA (sgRNA), which mediates recognition of a specific DNA sequence [19]. RNA-guided nuclease Cas9 was, therefore, converted to a sequence-specific DNA-binding platform, “deactivated Cas9” (dCas9), by inactivation of its two nuclease domains [19]. dCas9 can be fused to a trans-activation domain, such as VP64, to function as a transcriptional activator [20,21]. Further development brought more complex systems where the basic dCas9-VP64 module was extended by modifying the sgRNA structure such that sgRNA loops protruding from the ribonucleoprotein complex could carry sequences serving as binding platforms for additional transcription modulating proteins [22,23].

To test whether dCas9-mediated transcriptional activation could induce Dicer^O^ expression from the *Dicer* gene in mouse embryonic stem cells and fibroblasts, we examined several different dCas9-transcriptional activation designs and obtained the best Dicer^O^ activation with a system consisting of dCas9-VP64 enhanced with an MS2-p65-HSF1 module originally developed by Konermann et al. [23]. In this system, sgRNA loops protruding from the ribonucleoprotein complex carry a minimal hairpin aptamer, which is bound by the MS2 protein that can be fused with an additional factor enhancing transcription [22,23]. We report that we successfully induced robust Dicer^O^ protein expression and observed limited RNAi activity in mouse embryonic stem cells, but not in NIH 3T3 fibroblasts. Our results suggest that induction of RNAi in mouse cells may be possible by inducing Dicer^O^ expression, but achieving robust RNAi activity requires further optimization of Dicer^O^ induction, presumably combined with a strategy reducing the inhibitory effects of innate immunity factors.

## 2. Materials and Methods

### 2.1. Plasmids

Catalytically deactivated Cas9 (dCas) was constructed from a pSpCas9n^D10A^ plasmid (PX460; Addgene #48873) by mutating His to Ala at position 840 using Q5 site-directed mutagenesis kit (NEB) according to the manufacturer’s instructions. dCas9^D10A/H480A^ was transferred into the pPuro backbone containing the puromycin resistance gene. VP64 was PCR amplified from a pTALE-VP64-EGFP plasmid [14] and inserted into pPuro_dCas9 together with a flexible linker.

Different dCas9 plasmid variants were constructed by stepwise cloning of the following components: (1) Capsid assembly-defective MS2 coat protein variant dlFG lacking the FG loop [24] cloned in the form of a linked dimer [25] to augment dimerization. (2) Mouse p65 activator was PCR amplified from mouse genomic DNA (primers Fwd: 5′-CCATCAGGGCAGATCTCAAACCAGG and Rev: 5′- GGAGCTGATCTGACTCAAAAGAGC). (3) Human HSF1 was PCR amplified from HeLa cell cDNA (primers Fwd: 5′-GGCTTCAGCGTGGACACCAGTGCC and Rev: 5′-TCAGGAGACAGTGGGGTCCTTGGCTTTGG). (4) p300 HAT activation domain was PCR amplified from HeLa cell cDNA (primers Fwd: 5′-CCATTTTCAAACCAGAAGAACTACGAC and Rev: 5′-GTCCTGGCTCTGCGTGTGCAGCTC). When a single plasmid construct was employed, dCas9-VP64 was linked with the enhancing protein via the T2A self-cleavage peptide [23].

The sgRNA backbone containing MS2-binding sites (sgRNA^2.0^) was designed according to [23] and cloned under the U6 promoter. Desired targets were cloned by BsmBI restriction enzyme: Dcr_MT1 (-372): 5′-caccGAACAAATGGCTGCTGAA; Dcr_MT2 (-188): 5′-caccGTCAGTCATCTGAGGGAA; Dcr_MT3 (-78): 5′-caccGGCCCAACCCACTGTGGG; Dcr_MT4 (-162R): 5′-caccGAAGTACGTTCTCTATTG; Dcr_MT5 (-249R): 5′-caccGAGCATCACCCTCACTGA; Dcr_MT6 (-34R): 5′-caccGCTTTCTTAATAGAACCC; Dcr_MT7 (-59): 5′-caccGGGGCCATCCCTGGACTG; Dcr_MT8 (-269R): 5′-caccGTGGCAGTAACCCATTTG; Oct4-1 (-534): 5′-caccGGTCTCTGGGGACATATC ; Oct4-2 (-453): 5′-caccGCTGTCTTGTCCTGGCCT ; Oct4-3 (-216): 5′-caccGAGGTGTCCGGTGACCCA ; Oct4-6 (-170): 5′-caccGAAAATGAAGGCCTCCTG; Oct4-7 (-50): 5′-caccGCTCCTCCACCCACCCAG ; Oct4-4 (-275R): 5′-caccGTTGGCACTGCACCCTCT; Oct4-5 (-405R): 5′-caccGTCTAGAGTCCTAGATAT; Oct4-8 (-114R): 5′-caccGTCTTCCAGACGGAGGTT. The mouse *Dicer1* intron 6 region containing the MTC element was PCR amplified from mouse genomic DNA (Fwd: 5′-AAGCTTCTCGAGCCACCTTCAGTGAGGGTG and Rev: 5′-AAGCTTGTATGTCCTTTACACTGATTAAGC) and cloned into pGL4.10 plasmid (Promega; for simplicity referred to as FL) digested with HindIII. The mouse *Oct-4* (*Pou5f1*) promoter was PCR amplified from mouse genomic DNA (Fwd: 5′-CCATGGTGTAGAGCCTCTAAACTCTGGAGG and Rev: 5′-CCATGGGGAAGGTGGGCACCCCGAGCCGG) and cloned into pGL4.10 plasmid digested with NcoI. *Renilla* luciferase-expressing plasmid (Promega; for simplicity referred to as RL) was used for normalization.

An overview of different construct combinations used in the five characterized versions of the transcription-acting systems is provided in Table 1.

### 2.2. Cell Culture and Transfection

Mouse 3T3 cells were maintained in DMEM (Sigma-Aldrich, USA) supplemented with 10% fetal calf serum (Sigma-Aldrich, USA), penicillin (100 U/mL; Invitrogen, USA), and streptomycin (100 μg/mL; Invitrogen, USA) at 37 °C and 5% CO_2_ atmosphere. Mouse ESCs were cultured in 2i-LIF media: DMEM supplemented with 15% fetal calf serum, 1× L-glutamine (Invitrogen, USA), 1× nonessential amino acids (Invitrogen, USA), 50 μM β-mercaptoethanol (Gibco, Thermo Fisher Scientific, USA), 1000 U/mL LIF (Merck Millipore, Germany), 1 μM PD0325901, 3 μM CHIR99021, penicillin (100 U/mL), and streptomycin (100 μg/mL). For transfection, the cells were plated on a 24-well plate, grown to 50% density, and transfected using the TurboFect in vitro transfection reagent or lipofectamine 3000 (Thermo Fisher Scientific, USA) according to the manufacturer’s protocol. The cells were co-transfected with 50 ng per well of each FL and RL reporter plasmids and 250 ng per well of a dsRNA-expressing plasmid and, eventually, 250 ng per well of a plasmid expressing the tested factor. The total amount of transfected DNA was kept constant (600 ng/well) using a pBluescript plasmid. The cells were collected for analysis 48 h post-transfection.

### 2.3. Luciferase Assay

Dual-luciferase activity was measured according to [26] with some modifications as described previously [12]. Briefly, the cells were washed with PBS and lysed in PPTB lysis buffer (0.2% vol/vol Triton X-100 in 100 mM potassium phosphate buffer, pH 7.8). Aliquots of 3–5-μL were used for measurement in 96-well plates using a Modulus microplate multimode reader (Turner Biosystems, USA). First, firefly luciferase activity was measured by adding 50 μL substrate (20 mM tricine, 1.07 mM (MgCO_3_)_4_·Mg(OH)_2_·5 H_2_O, 2.67 mM MgSO_4_, 0.1 mM EDTA, 33.3 mM DTT, 0.27 mM Coenzyme A, 0.53 mM ATP, and 0.47 mM D-luciferin, pH 7.8) and the signal was integrated for 10 s after a 2 s delay. The signal was quenched by adding 50 μL *Renilla* substrate (25 mM Na_4_PP_i_, 10 mM sodium acetate, 15 mM EDTA, 500 mM Na_2_SO_4_, 500 mM NaCl, 1.3 mM NaN_3_, and 4 μM coelenterazine, pH 5.0) and *Renilla* luciferase activity was measured for 10 s after a 2 s delay.

### 2.4. Western Blotting

Cells were grown in 6-well plates. Before collection, the cells were washed with PBS and lysed in RIPA buffer (50 mM Tris, pH 7.5, 150 mM NaCl, 1 mM EDTA, 1 mM EGTA, 1% NP-40 (Igepal CA-630), 0.5% Na-deoxycholate, 0.1% SDS) supplemented with 2× protease inhibitor cocktail set (Merck Millipore, Germany). Proteins were separated in 5% (for Dicer detection) or 10% (for Tubulin detection) polyacrylamide gel and transferred to a PVFD membrane (Merck Millipore, Germany). Anti-Dicer (#349; [27]) and anti-Tubulin (#T6074, 1:5000; Sigma-Aldrich, USA) primary antibodies and HRP-conjugated secondary antibodies (1:50,000) were used for signal detection with SuperSignal West Femto chemiluminescent substrate (Pierce, Thermo Fisher Scientific, USA ).

### 2.5. Native Chromatin Immunoprecipitation

Cells were washed in PBS and resuspended in nuclei preparation buffer I (0.3 M sucrose, 60 mM KCl, 15 mM NaCl, 5 mM MgCl_2_, 0.1 mM EGTA, 0.5 mM DTT, 0.2% NP-40, 15 mM Tris/HCl, pH 7.5, supplemented with protease inhibitors) and nuclei were released by passing three times through a 21G needle.

The sample was placed over nuclei preparation buffer III (1.2 M sucrose, 60 mM KCl, 15 mM NaCl, 5 mM MgCl_2_, 0.1 mM EGTA, 0.5 mM DTT, 15 mM Tris/HCl, pH 7.5) and centrifuged at ~2000× *g* for 15–20 min. The pellet (containing purified nuclei) was resuspended in MNase digestion buffer (0.32 M sucrose, 1 mM CaCl_2_, 4 mM MgCl_2_, 50 mM Tris/HCl, pH 7.5) to bring final DNA concentration to 1000 mg/mL. Aliquots of 500 µg chromatin DNA were digested by 5 U MNase (micrococcal nuclease) for 5 min at 37 °C. Digestion was stopped by adding EDTA to the final concentration >20 mM. Supernatant containing mono- and oligo-nucleosomes was used for immunoprecipitation: 30 µg chromatin DNA and 5 µg of primary antibody (anti-H3K9me2 (Abcam #ab1220), anti-H3K9me3 (Abcam #ab8898), anti-H3K9ac (Abcam #ab4441) or control rabbit IgG (Millipore #PP64B)) were used per sample in 700 µL NChIP incubation buffer (50 mM NaCl, 5 mM EDTA, 50 mM Tris/HCl, pH 7.5). Pre-blocked Dynabeads protein G (Invitrogen) or protein A-sepharose 4B, fast flow from *Staphylococcus aureus* (Sigma) were used for pull-down. Beads were washed in wash buffer (10 mM EDTA, 50 mM Tris/HCl, pH 7.5) containing increasing NaCl concentrations (75, 125, 175, and 300 mM; two washes each).

DNA elution was performed in “PCR-friendly” elution buffer (50 mM KCl, 0.45% NP-40, 0.45% Tween-20, 0.01% gelatin, 10 mM Tris/HCl, pH 8.0) supplemented with 1 µL proteinase K (Fermentas) added to the beads pellet. Samples were incubated for 1 h at 55 °C followed by proteinase K inactivation for 10 min at 95 °C. Supernatants were used directly for qPCR analysis using the following primers: Dcr_int2shortR 5′-TGCCCTACAGGTGTGTCTGT; Dcr_int2midR 5′-CTGTGTAGAGGTGTCTGTTTCCA; Dcr_int2F 5′-GCAAAGACCAGCTCCAGCCAT; mDcr_AltE_Fwd 5′-CTCTGCCTTCAGGTTCTGACTTCC.

### 2.6. qPCR Analyses

Total RNA was isolated using RNeasy MINI kit (Qiagen), and 1 µg amount was used for the reverse transcription (RT) reaction using RevertAid premium (Fermentas) in a 30 µL volume. A 0.5 µL aliquot was used per qPCR reaction. Maxima SYBR Green qPCR master mix (Thermo Scientific) was ge C_T_ values of the technical replicates were normalized toused for qPCR. qPCR was performed in technical triplicates for each biological sample. Avera housekeeping genes *Hprt* and *B2MG* using the ΔΔC_T_ method [28]. Primers used for qPCR were as follows: *Pou5f1*: mPou5f1_qPCR_Fwd 5′-GTTGGAGAAGGTGGAACCAA and mPou5f1_qPCR_Rev 5′-GCAAACTGTTCTAGCTCCTTCTG; Dicer_MT (1): mDcr_AltE_Fwd 5′-CTCTGCCTTCAGGTTCTGACTTCC and mDcr_E7_Rev 5′-GCAATCTCTCATACAGCCCACTTC; Dicer_MT (2): mDcr_AltE_Fwd 5′-CTCTGCCTTCAGGTTCTGACTTCC and mDcr_E7_Rev2 5′-CAGTCTACCACAATCTCACAAGGC; *Hprt1*: mHPRT.1.457_Fwd 5′-GCTACTGTAATGATCAGTCAACGG and mHPRT.1.670_Rev 5′-CTGTATCCAACACTTCGAGAGGTC; *B2m*: mβ2-MG.1.342_Fwd 5′- GCAGAGTTAAGCATGCCAGTATGG and mβ2-MG.1.514_Rev 5′-CATTGCTATTTCTTTCTGCGTGC.

## 3. Results and Discussion

As mentioned above, Dicer^O^ is not expressed in other mouse cells than oocytes. Using chromatin immunoprecipitation, we examined histone modifications of the *Dicer^O^* promoter in NIH 3T3 cells to test whether *Dicer^O^* silencing involves more reversible facultative heterochromatin or more stable constitutive heterochromatin, which may be more difficult to reverse into euchromatin. Chromatin immunoprecipitation showed the presence of facultative heterochromatin mark H3K9me2, but not the constitutive heterochromatin mark H3K9me3 at the MTC LTR insertion controlling *Dicer^O^* expression (Figure 1C). Regarding DNA methylation, the MTC LTR promoter is a CpG poor promoter where the presence of DNA methylation may not be a significant silencing factor [29].

To develop transcriptional reactivation of Dicer^O^ in somatic cells, we tested several different designs built on the CRISPR dCas9 system recruiting transcriptional transactivator(s) into the *Dicer^O^* promoter locus. We started with a basic transcriptional activating system where dCas9 is fused with the VP64 transcriptional transactivator [20,21] and is guided by unmodified sgRNA (Figure 2A).

For initial testing of the CRISPR-driven transcriptional reactivation, we produced stable NIH 3T3 cell lines expressing dCas9-VP64 or dCas9-VP64-EGFP [14] transcriptional activators. This allowed us to reduce the number of transfected plasmids and ensure that all cells would have a constant amount of dCas9-VP64 expression. Cells stably expressing dCas9-VP64 were then co-transfected with an expression vector for sgRNA to target dCas9-VP64 into the promoter of interest cloned into a firefly luciferase reporter and an SV40 promoter-driven *Renilla* luciferase reporter allowing for normalization of the luciferase activity (Figure 2B). Two different firefly reporters were produced to examine the efficiency of the dCas9 activation system, *Pou5f1* (*Oct-4*) promoter and MTC-derived promoter for Dicer^O^; for each promoter, we designed and tested several sgRNAs.

Experiments with the *Pou5f1* promoter-reporter and a set of sgRNAs have shown that the system can induce reporter expression (Figure 2C). However, individual sgRNAs had just marginal effects. At the same time, strong transcriptional activation was observed when several sgRNAs were combined (Figure 2C). Because the NIH 3T3 line expressing the dCas-VP64-EGFP fusion yielded an order of magnitude lower transcriptional activation estimated by the luciferase assay, we omitted the dCas-VP64-EGFP fusion from subsequent experiments and continued with dCas-VP64 (denoted version 1). When testing MTC promoter activation, we also observed that individual sgRNAs had a much lower effect than some of their combinations. The most powerful combination was observed with a mixture of all four tested sgRNAs (2–5) that could bind the promoter cloned into the firefly luciferase reporter (Figure 2D). When examining transcriptional activation of the endogenous MTC promoter of *Dicer^O^* in 3T3 cells, the system yielded up to 90-fold transcriptional activation of *Dicer^O^* over the background level (Figure 2E). Of note is that qPCR analysis of the endogenous promoter and luciferase reporter gave results differing for some sgRNAs and their combinations (e.g., sgRNA 4 vs. sgRNA 5 in Figure 2D,E). This may reflect different accessibility of the plasmid reporter and the genomic locus.

To test whether we could further enhance transcriptional activation effects, we employed a published sgRNA modification with MS2 binding sites, which allows binding of exposed sgRNA loops to the MS2 domain fused with additional transcriptional activators [23]. We tested three variants of the MS2 system: (1) an MS2-p65-HSF1 fusion construct with enhanced dimerization MS2 variant (MS2^dlFG^ [24]), which was encoded in a single construct together with dCas9-VP64 via T2A self-cleavage peptide as described previously [23]. We denoted this system as of version 2 (Figure 3A); (2) modification of the MS2 component by using a covalently linked MS2-dimer (dimMS2^dlFG^-p65-HSF1 [25]) in a single construct with dCas9-VP64, denoted version 3; and (3) use of separate constructs expressing dCas9-VP64 and dimMS2^dlFG^-p65-HSF1, denoted version 4 (Figure 3B). For each of the versions, we produced stable 3T3 lines and examined *Pou5f1* and MTC reporter expression as described previously (Figure 2B). A detailed overview of the variants of dCas9-VP64 systems is provided in Table 1. Expression of dCas9 and enhancing factors in stable cell lines is provided in Appendix A.

*Pou5f1* reporters (Figure 3C) revealed the best enhancing effects of version 4 of the transcriptional activation system; otherwise, the effects were comparable to the Cas9-VP64 version without MS2-based enhancement (Figure 2C). Importantly, transcriptional activation of the endogenous MTC promoter of Dicer^O^ in 3T3 cells with version 4 was much stronger than with version 3 (Figure 3D). When we used transient transfection and compared version 1 and version 4, version 4 was clearly superior to version 1, suggesting that modified sgRNA and dimMS2^dlFG^-p65-HSF1 indeed have an enhancing effect (Figure 3E). In the same experiment, we also examined alternative MS2-based enhancement employing dimMS2^dlFG^-p300 acetyltransferase fusion [22], denoted version 5, but version 4 using separate constructs expressing dCas9-VP64 and dimMS2^dlFG^-p65-HSF1 was clearly superior (Figure 3E). We thus selected version 4 for analysis of the production of the Dicer^O^ protein.

To analyze Dicer^O^ production’s activation, we used stable cell lines expressing version 4 (Figure 4A) and transient transfection of sgRNA-expressing constructs (Figure 4B). When deploying the system into embryonic stem cells (ESCs), we were able to detect robust transcriptional activation and production of Dicer^O^ protein, whose abundance in the lysate of transiently transfected cells was comparable to the endogenous Dicer protein (Figure 4C). Although we achieved clear transcriptional activation of *Dicer^O^* in NIH 3T3 cells, the *Dicer^O^* transcript levels were approximately a hundredfold less induced than in ESCs, and the truncated protein could not be detected by Western blotting (Figure 4D). Because of this result, we tested the induction of RNAi activity in ESCs.

To test the RNAi activity (Figure 4E), we co-transfected ESCs stably expressing version 4 with a combination of four sgRNAs that yielded the most of the Dicer^O^ protein and our established RNAi assay system described in detail previously [12]. Briefly, the RNAi assay system has three plasmid components: (1) a targeted *Renilla* luciferase reporter with a cognate *Mos* sequence in the 3′ UTR, (2) a non-targeted firefly luciferase reporter, and (3) a plasmid expressing long dsRNA with the *Mos* sequence in the form of a long hairpin (MosIR). As a control for nonspecific dsRNA effects, we used unrelated dsRNA-expressing plasmids (Lin28IR and Elavl2IR) instead of MosIR. As negative control to MosIR, we used a MosMos construct, where *Mos* sequences are oriented head-to-tail. Hence the plasmid has the same sequence as MosIR but does not produce dsRNA. Western blot analysis again showed good induction of Dicer^O^ expression and a marked reduction of the targeted *Renilla* reporter in the presence of MosIR (Figure 4E). The average *Renilla* reporter reduction was ~30% (*p*-value > 0.05), which was comparable to our previous experiments with cells that have an intact protein kinase R gene encoding one of the dsRNA-sensing components of the interferon response [12,14].

To sum up, we tested five different versions of the dCas9-VP64 transcriptional activation system, out of which we selected dCas9-VP64 combined with sgRNA carrying MS2-binding sites and a dimMS2^dlFG^-p65-HSF1 fusion protein. Our work shows the extent of optimization needed to induce robust transcriptional activation (even if only in ESCs). It underscores the importance of setting up a good testing system for multiple sgRNAs and their combinations. We opted for producing stable lines expressing dCas9 and the enhancing factor. These cell lines represent one of the valuable contributions of this work as they can be used to program dCas9 with sgRNAs to influence gene expression in ESCs and 3T3 cells.

Using this three-component system and a combination of four sgRNAs targeting the MTC promoter of *Dicer^O^* isoform, we achieved robust Dicer^O^ protein expression in ESCs comparable to full-length Dicer expression. In fact, Dicer^O^ protein expression in cells transfected with sgRNAs likely exceeded that of endogenous full-length Dicer in those cells because in Western blots in Figure 4C,E, the Dicer^O^ signal comes from fewer cells than the full-length Dicer protein detected above it. However, despite the level of Dicer^O^ expression, the RNAi activity assessed by a reporter assay was minor, if any, suggesting that the sole expression of Dicer^O^ is not sufficient to bring robust RNAi. On the other hand, this observation is consistent with other data showing that RNAi activity is low in the presence of specific innate immunity factors [9,10,11,12].

Unfortunately, we did not succeed in inducing a similar expression of Dicer^O^ in NIH 3T3 cells. This result seems to be caused by much lower transcriptional activation than in ESCs and could be influenced by different chromatin organization in ESCs and NIH 3T3 cells, which are derived from fibroblasts [30]. ESCs have a relatively open chromatin structure supporting higher transcription and reduced heterochromatin signature [31,32]. Thus, ESCs may better respond to transcriptional reactivation of *Dicer^O^* than NIH 3T3 cells. The system with histone acetyltransferase p300 (version 5) did not have an additive effect on top of VP64 (Figure 3E), suggesting that targeting acetylation to the locus does not sufficiently enhance transcriptional activation of the MTC promoter. Perhaps a histone lysine demethylase targeted to the locus that would reduce H3K9me2 could make the transcriptional activation of *Dicer^O^* more efficient.

In conclusion, we showed that reactivation of Dicer^O^ via a dCas9 system is not a viable strategy to induce a robust canonical RNAi pathway in cultured mouse cells. While rather negative, our results are useful for understanding the functional limits of the endogenous RNAi and long dsRNA processing capacity furnished by the expression of Dicer^O^ from its endogenous locus. Although this time we did not observe robust RNAi, it is likely that more sensitive methods, such as RNA sequencing of small RNAs, would reveal the impact of expressed truncated Dicer on the long dsRNA metabolism. In our previous work in 3T3 and ESCs, we observed a robust increase in siRNA biogenesis with ectopic expression of Dicer^O^ but rarely robust RNAi [12]. Our work offers another tool for studying the consequences of Dicer^O^ expression in mouse cells and investigating whether low-activity RNAi would have any measurable impact on cell physiology, especially dsRNA metabolism, miRNA pathway, the sensitivity of innate immunity pathways to dsRNA, and viral resistance.

## Figures and Tables

**Figure 1 genes-12-00540-f001:**
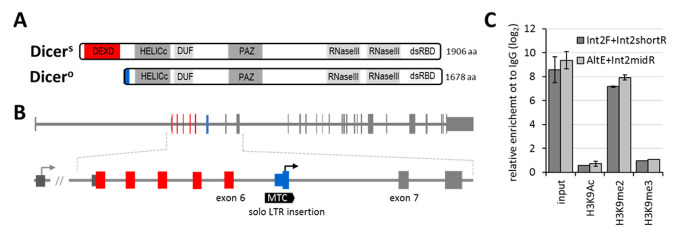
(**A**) Schemes of murine oocyte-specific truncated Dicer protein (Dicer^O^) and full-length Dicer (Dicer^S^, S for somatic). (**B**) Genomic organization of the *Dicer1* gene indicating the position of the alternative promoter and the first exon encoding Dicer^O^. The alternative promoter and the first exon reside in MTC LTR insertion between exons 6 and 7, and its structure and evolutionary history have been described elsewhere [15]. (**C**) Native chromatin immunoprecipitation analysis of H3K9 modifications in the MTC Dicer^O^ promoter region in NIH 3T3 cells. Relative enrichment analyzed by two primer pairs is depicted as log_2_-fold enrichment calculated to IgG.

**Figure 2 genes-12-00540-f002:**
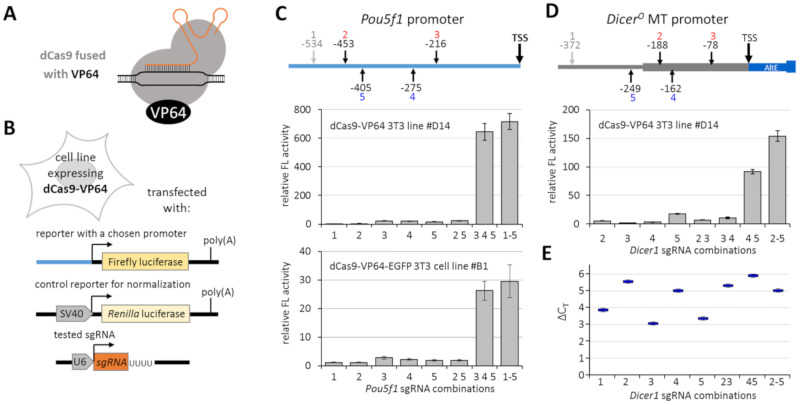
Transcriptional activation using a basic dCas9-VP64 system. (**A**) Schematic depiction of the basic dCas9-VP64 system version 1 composted of dCas9 fused with VP64 and a sgRNA. (**B**) Schematic depiction of the testing strategy built on NIH 3T3 cell lines stably expressing dCas9-VP64 fusions and transfected with plasmids expressing sgRNAs, a firefly reporter driven by a tested promoter, and a *Renilla* luciferase reporter as a control for normalization of transcriptional activation of the tested promoter. (**C**) Transcriptional activation of *Pou5f1* promoter-reporter. The scheme depicts positions of sgRNAs relative to the *Pou5f1* transcription start. sgRNA 1 was outside of the promoter region cloned into the firefly reporter, hence could serve as a negative control. The upper graph depicts results with dCas9-VP64, the lower one with dCas9-VP64-EGFP. Shown is the firefly luciferase activity normalized to *Renilla* luciferase; reporter activity in the absence of sgRNA was set to 1. (**D**) Transcriptional activation of the MTC promoter-reporter. As in (**C**), the scheme depicts the position of sgRNAs relative to the transcription start site, and sgRNA 1 was outside of the cloned promoter region. (**E**) Transcriptional activation of the endogenous MTC promoter in NIH 3T3 cells. The graph depicts relative expression calculated by the ∆∆C_T_ method; negligible basal expression detected in NIH 3T3 cells was set to 1. All error bars = standard deviation.

**Figure 3 genes-12-00540-f003:**
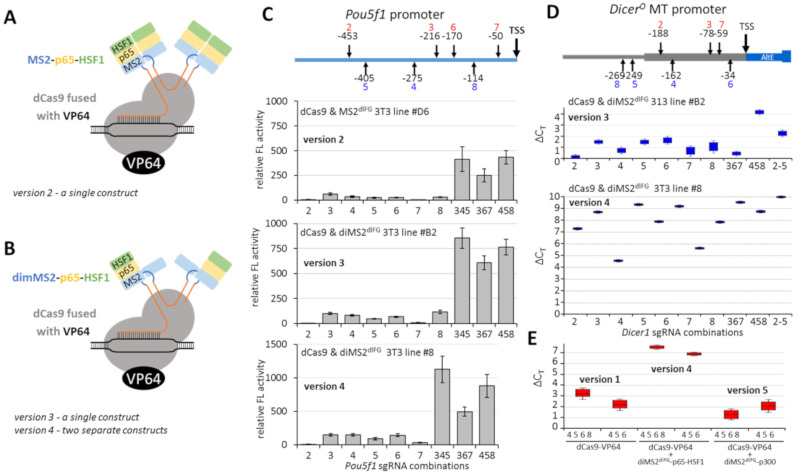
Transcriptional activation using dCas9-VP64 enhanced with MS2 systems. (**A**) Schematic depiction of version 2 based on dCas9-VP64, modified sgRNA with MS2-binding sequence (blue) and MS2 domain fused with p65 and HSF1. The scheme indicates that the MS2 fusion construct is dimerizing on the RNA sequence. (**B**) Schematic depiction of versions 3 and 4. The difference from (A) lies in MS2 being encoded as a covalently linked duplex where one MS2 is fused with p65 and HSF1 (dimMS2-p65-HSF1). Versions 3 and 4 differ in the organization of expression. Version 3 expresses dCas9-VP64 and dimMS2-p65-HSF1 from a single construct, version 4 from separate constructs. (**C**) Transcriptional activation of *Pou5f1* promoter-reporter with versions 2, 3, 4. As in Figure 1C, shown are relative firefly luciferase activities normalized to the *Renilla* control reporter; firefly reporter activity in the absence of sgRNA was set to 1. (**D**) Transcriptional activation of the endogenous MTC promoter in NIH 3T3 cells. The graph depicts relative expression calculated by the ∆∆C_T_ method. (**E**) Comparison of transcriptional activation of the endogenous MTC promoter in NIH 3T3 cells of version 4 with version 1 (Figure 2) and version 5, which uses dimMS2-p300 instead of dimMS2-p65-HSF1. The graph depicts relative expression calculated by the ∆∆C_T_ method. All error bars = standard deviation.

**Figure 4 genes-12-00540-f004:**
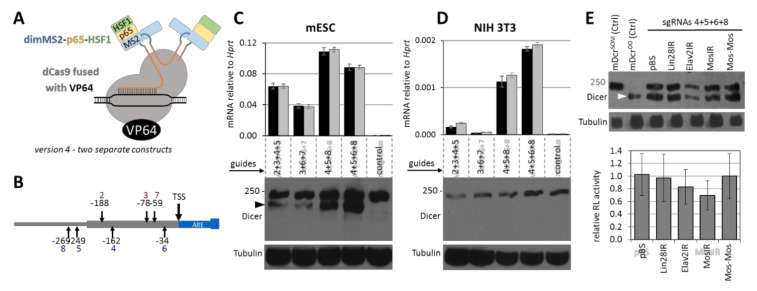
Transcriptional activation of Dicer^O^. (**A**) Schematic depiction of the finally chosen version 4. (**B**) Schematic depiction of the position of sgRNAs targeting the MTC promoter. (**C**) Activation of Dicer^O^ with shRNA combinations in ESCs analyzed by qPCR and Western blot. Dicer^O^ is visible as the lower band detected by the anti-Dicer antibody (depicted by black arrowhead). (**D**) Activation of Dicer^O^ with shRNA combinations in NIH 3T3 cells analyzed by qPCR and Western blot. Dicer^O^ was not detected. (**E**) Analysis of RNAi effects in ESCs where Dicer^O^ expression was induced with sgRNAs. Western blot depicts activation of Dicer^O^ in ESCs (depicted by white arrowhead). The first two Western blot lanes show control ESCs expressing either the full-length Dicer^S^ or truncated Dicer^O^ [14]. The graph depicts the relative activity of a *Renilla* luciferase reporter carrying a cognate *Mos* sequence as described previously [12]. Briefly, the *Renilla-Mos* reporter is co-transfected with a non-targeted firefly luciferase reporter and a reporter expressing dsRNA from an inverted repeat (IR). pBS and MosMos are controls not expressing dsRNA. All error bars = standard deviation.

**Table 1 genes-12-00540-t001:** Overview of versions of transcription activation systems used in the study.

System Version	dCas9 Version	sgRNA Version	Modulating Co-Factor	Comment
Version 1	HA-dCas9^D10A/H480A^-VP64 (HA-tagged)	U6-driven sgRNA	none	
Version 2	HA-dCas9^D10A/H480A^-VP64 (HA-tagged)	U6-driven sgRNA with two MS2-binding sites	MS2^dlFG^-p65-HSF1	dCas9 and modulator in a single construct (linked with T2A)
Version 3	HA-dCas9^D10A/H480A^-VP64 (HA-tagged)	U6-driven sgRNA with two MS2-binding sites	dimMS2^dlFG^-p65-HSF1	dCas9 and modulator in a single construct (linked with T2A)
Version 4	HA-dCas9^D10A/H480A^-VP64 (HA-tagged)	U6-driven sgRNA with two MS2-binding sites	dimMS2^dlFG^-p65-HSF1-HA (HA-tagged)	dCas9 and modulator in separate plasmids
Version 5	HA-dCas9^D10A/H480A^-VP64 (HA-tagged)	U6-driven sgRNA with two MS2-binding sites	dimMS2^dlFG^-p300-HA (HA-tagged)	dCas9 and modulator in separate plasmids

## Data Availability

Data is contained within the article or supplementary material.

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
