# Peer review of "CRISPR-Induced Expression of N-Terminally Truncated Dicer in Mouse Cells"

_genes, 2021, doi:10.3390/genes12040540_

Round 1

Reviewer 1 Report

The authors of this manuscript seek to create a stable expression of Dicero through the transfection of CRISPR components targeting the MTC region of Dicer into two mouse cell lines. From their results they are able to derive a combination of sgRNAs and dCas9 constructs to result in high Dicero production in mice stem cells. However, this Dicero production does not result in high RNAi activity for knockdown of Mos. While the claims of this paper are supported by the presented results, this reviewer has concerns of the composition of the manuscript that needs to be addressed prior to publication.

  1. For broad audience understanding, need to provide more details on the CRISPR system and the modifications used. In its current state it is difficult for readers to understand each of the activators, components, and sgRNA locations without having to do further reading.
  2. It is not clear why the authors are trying to activate Dicero. Overall the authors need to establish a goal and provide an explanation of what the purpose of expressing Dicero in cells would do. I am guessing that this would be a complement to other delivered RNAi sequences for therapeutics and enhance RNAi efficacy. But this is not explicitly described.
  3. The authors should provide a prospective of how this can be translated into human, can Dicero exist in humans or only in mice? What would be a comparable study in human cell and what implications can such technology cause.
  4. Additionally the authors transfect the CRISPR related plasmids into the cells, but this would not be practical for in vivo studies. While this is a proof-of-concept manuscript the authors should at least provide a perspective in the discussion of how this technology could be delivered for in vivo applications.
  5. Minor Points

-In Fig. 2D, 3C, and 3D no data is shown for position 1 as negative control but is shown in Fig. 2C. This data should be included.

-Please provide an explanation for the difference in Firefly expression (Fig. 2D) over qPCR data (Fig. 3E) for samples 3, 5, and 23. This review does not understand why little effect is seen in the luciferase test but high expression change in the qPCR test. Similarly in Fig, 3D v3.1 for samples 2, 5, 7.

Reviewer 2 Report

This is an informative and dense paper that provided useful tools for future research on CRISPR, RNAi, and ESC. I recommend making the following changes:

  1. Briefly summarize the findings in the Abstract.
  2. To test what kind of heterochromatin is involved in Dicero silencing, why not use CHIP-seq directly on NIH 3T3 cells? MTC LTR insertion might change histone modifications controlling Dicero expression; therefore, doing CHIP after the insertion might not reveal the original state of histone modifications. A comparison of epigenetic states before and after the insertion could generate more insights.
  3. Add more details and background information in Introduction and when explaining experiment rationales. For example, the experiment described in lines 266-268 would benefit from more specifics, especially if the audience has not read the previous paper.
  4. In line 272-273, “we could achieved” should be changed to “we could achieve” or something else that is grammatically correct. In line 276, “then” should be “than”. I suggest systematically editing the paper to eliminate grammatical errors and improve clarity.
  5. Enrich the discussion section by adding implications for future research and for the discovery of new therapies. 

Reviewer 3 Report

Malik and Svodoba et al highlights the role of RNAi machinery (in this case- Dicer) which is relatively less understood in maintaining mammalian heterochromatin. Using CRISPR based methods they activate Dicer0 in ESCs and show that CRISPR activated Dicer can mediate RNAi.  This presents an interesting piece of work with potential for improving the CRISPR trancription activation constructs. CRISPR based epigenetic targeting is already well established and therefore is not novel.

Specific comments

1) Please refrain from using unscientific nomenclature/acronym for various constructs like version1.0 .2.0 etc . This is not a software but different constructs. it is very hard to search the text for what version 3 or 4 is . therefore better to have a long name that rather uninformative nomenclature.

2)There has to data showing expression levels of dCas9 VP64 in version 1,2,3 using western blot and that the cells are stably expressing the modified cas9. There is no data for that. Detailed data for the validation of the various constructs generated must be shared as supplementary.

3) Authors must perform H3K4me3 and H3K9me2/3 ChIP qPCR for the Dicer0 locus in mESC and NIH3T3 to ascertain why in mESCs there is activation and not in the fibroblast. This will prove or disprove their hypothesis that the epigenetic landscape at the locus is different in the two cell types leading to variable results.

4) They can also add qPCR data for the innate immunity factors pathway genes to ascertain if that contributes to lack to activation in mESCs vs NIH 3T3

5) Authors should try to bring out and discuss in the discussion section what is the scientific importance of these results. what do we learn that we did not know.  what are the significance of these findings? Discussing or even comparing with some experimental data how the constructs generated in this paper provides more robust transcriptional activation compared to already reported constructs will be helpful.

6) Authors missed some more recent reviews on the mammalian heterochromatin eg Saha et al Febs J 2019, Iglesias 2017 eLife; Martienssen 2015, CSHL perspectives etc. 

Round 2

Reviewer 1 Report

The authors have addressed the concerns raised and provided a better broad understanding of the presented work with the added introduction, table, and conclusions. Their responses provide a better understanding to their presented work.

Reviewer 3 Report

Not enough data to support the conclusions.